# Acquisition, co-option, and duplication of the *rtx* toxin system and the emergence of virulence in *Kingella*

Daniel P. Morreale [1,2], Eric A. Porsch [2], Brad K. Kern[1,2], Joseph W. St. Geme III [1,2,4] & Paul J. Planet [1,2,3,4] ✉

The bacterial genus *Kingella* includes two pathogenic species, namely *Kingella kingae* and *Kingella negevensis*, as well as strictly commensal species. Both *K. kingae* and *K. negevensis* secrete a toxin called RtxA that is absent in the commensal species. Here we present a phylogenomic study of the genus *Kingella*, including new genomic sequences for 88 clinical isolates, genotyping of another 131 global isolates, and analysis of 52 available genomes. The phylogenetic evidence supports that the toxin-encoding operon *rtxCA* was acquired by a common ancestor of the pathogenic *Kingella* species, and that a preexisting type-I secretion system was co-opted for toxin export. Subsequent genomic reorganization distributed the toxin machinery across two loci, with 30-35% of *K. kingae* strains containing two copies of the *rtxA* toxin gene. The *rtxA* duplication is largely clonal and is associated with invasive disease. Assays with isogenic strains show that a single copy of *rtxA* is associated with reduced cytotoxicity in vitro. Thus, our study identifies key steps in the evolutionary transition from commensal to pathogen, including horizontal gene transfer, co-option of an existing secretion system, and gene duplication.

*K ingella kingae* is a Gram-negative coccobacillus that was originally thought to be a rare cause of human disease and is now known to be an important pathogen, especially in young children. *K. kingae* initiates infection by colonizing the oropharynx, where it can persist for weeks to months without causing symptoms[1–5]. On occasion, *K. kingae* translocates across the oropharyngeal epithelial barrier, enters the bloodstream, and disseminates to distant sites such as bones, joints, and the endocardium to cause disease, generally in the setting of viral infection of the oral cavity or upper respiratory tract[1]. In many parts of the world, *K. kingae* has emerged as the most common cause of septic arthritis in children between 6 and 48 months of age[6].

The development of invasive disease by *K. kingae* is facilitated by several virulence factors, including an RTX family toxin, type IV pili, an autotransporter adhesin, a lipopolysaccharide-associated exopolysaccharide, and a polysaccharide capsule[7]. Together, these factors are thought to allow *K. kingae* to adhere to and lyse host cells, resist complement-mediated killing and neutrophil phagocytosis in the bloodstream, and subvert neutrophil killing at the site of disease. Of the five species that comprise the *Kingella* genus, only *K. kingae* and *K. negevensis* have been repeatedly associated with invasive disease in immunocompetent individuals[5,8]. These two species are also distinguished from the rest of the genus by the presence of an RTX family toxin called RtxA, which is encoded by the *rtxA* gene and causes β-hemolysis and toxicity to a variety of human cell types[9]. In both *K. kingae* and *K. negevensis*, the toxin-associated genes are hypothesized to have been acquired as a part of a mobile genetic element, though no study has defined the evolutionary history of this locus in detail in diverse *K. kingae* isolates[9–11].

The RtxA toxin is highly conserved among *K. kingae* isolates and is required for morbidity and mortality in an animal model of invasive

[1]Perelman School of Medicine, University of Pennsylvania, Philadelphia, PA, USA. [2]Division of Infectious Diseases, Children's Hospital of Philadelphia, Philadelphia, PA, USA. [3]Comparative Genomics, American Museum of Natural History, New York, NY, USA. [4]These authors contributed equally: Joseph W. St Geme III, Paul J. Planet. ✉ e-mail: planetp@chop.edu

disease[12]. RtxA is modified by the RtxC acyltransferase and is secreted from the organism by a type-I secretion system (TISS) that is encoded by the products of the *rtxB*, *rtxD*, and *tolC* genes. The toxin-associated genes are present at two loci in *K. kingae*, including one locus that contains *rtxB*, *rtxD*, and *rtxC* and a second locus that contains *rtxC*, *rtxA*, and the *tolC* gene, which encodes the TolC outer membrane protein[10]. In at least one strain (KWG-1), there are two copies of both *rtxA* and *tolC*, with the second copy of these genes downstream of *rtxB* and *rtxD*[7,10,13]. The impact of this duplication on virulence and pathogenicity is unknown.

Prior studies have shown that certain clonal groups identified by pulsed-field gel electrophoresis (PFGE) or multilocus sequence typing (MLST) are associated with invasive disease. For example, 70% of *K. kingae* invasive isolates belong to PFGE groups B, H, K, N, and P (out of 32 total groups). PFGE group K has been associated with occult bacteremia, group N with skeletal infections, and group P with endocarditis[14]. Similarly, ST-6, ST-14, ST-23, ST-25, and ST-66 have been implicated in *K. kingae* outbreaks in daycare centers[3,15,16]. No detailed analysis has been conducted on *K. kingae* genomes to identify virulence-associated genes or genotypes in these sequence types.

In this study, we hypothesized that there are specific genomic features that are responsible for both the emergence of virulence in *K. kingae* and *K. negevensis* and the association of particular lineages with a greater likelihood of invasiveness. We found that the efficient

secretion of RtxA is the result of co-option of a preexisting type-I secretion system, a critical step in the evolution of *K. kingae* and *K. negevensis* virulence. Phylogenetic analysis of 88 *K. kingae* clinical isolates sequenced for this study along with all 52 publicly available *K. kingae* genomes allowed us to test for lineages and genomic features associated with invasive disease. We discovered that a duplication of *rtxA* is associated with invasive disease and is concentrated primarily in a single clade. Additional analysis established that a single copy of *rtxA* is associated with reduced virulence compared to two copies of *rtxA* in vitro.

## Results

### A common ancestor of pathogenic *Kingella* species acquired the RTX toxin

Consistent with existing literature, when the five *Kingella* species were assessed for β-hemolysis on BHI plates supplemented with 10% sheep blood, only *K. kingae* and *K. negevensis* were hemolytic (Fig. 1A), suggesting that the two pathogenic *Kingella* species produce the toxin and raising the possibility that the toxin gene was acquired by the common ancestor of these species. Indeed, the lack of *rtxA* genes detected in the commensal *Kingella* species has previously led investigators to suggest that *rtxA* was acquired horizontally by *K. kingae*[2,9,10]. To better understand the acquisition of the RtxA toxin in *K. kingae* and *K. negevensis*, we first needed to

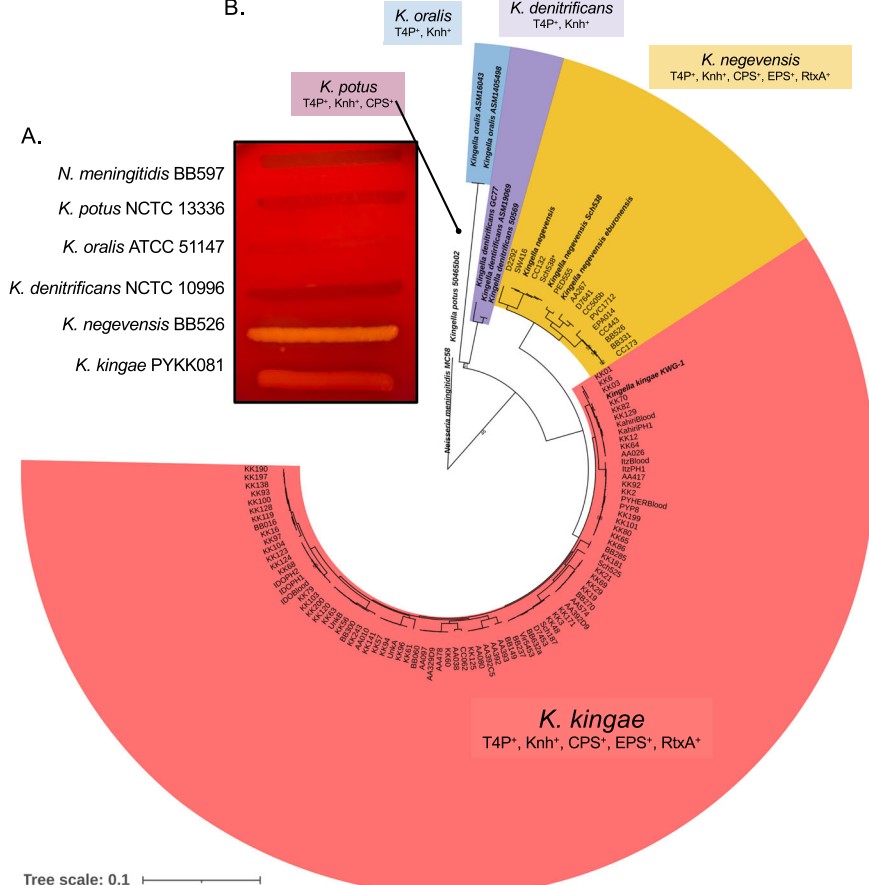

**Fig. 1 | Phylogenetic reconstruction of *Kingella* species. A** Representative strains of *N. meningitidis*, *K. potus*, *K. oralis*, *K. denitrificans*, *K. negevensis*, and *K. kingae* were grown on blood plates. Only *K. negevensis* and *K. kingae* produce any zone of hemolysis. **B** A maximum likelihood phylogeny of all bacterial strains sequenced in this study was reconstructed. To aid in resolving species boundaries, *N. meningitidis* MC58, *K. potus*, *K. oralis*, *K. denitrificans*, *K. negevensis*, and *K. kingae* KWG-1 were included from public databases as signposts. These isolates are denoted in bold and include genus and species names. The phylogeny is rooted by *N. meningitidis* MC58, and the branches and clades are colored by species. Bootstraps *N* = 100; bootstrap values less than 75 are indicated along the branches. The presence/absence of known virulence factors is indicated along with the species label. T4P: Type IV pili; Knh: Knh trimeric autotransporter; CPS: capsular polysaccharide; EPS: exopolysaccharide; RtxA: RtxA toxin gene. The Newick formatted tree is available in the source data files.

understand the genetic history of the toxin and associated machinery in the two pathogenic *Kingella* species.

To determine the relatedness of *Kingella* strains, a whole-genome phylogenetic analysis was performed based on 292,220 nucleotide sites (Fig. 1B). As expected, *K. negevensis* and *K. kingae* are sister taxa that constitute a single clade of pathogenic *Kingella* species, reinforcing the proposed acquisition of *rtxA* in the common ancestor of these species. Recent work from our group showed that *K. kingae* and *K. negevensis* diverged from *Alysiella* and *Simonsiella* more recently than they diverged from *K. denitrificans*[17]. Genome analyses showed no evidence of *rtxC* or *rtxA* genes in any of these closely related taxa.

### *K. negevensis* isolates encode the RTX machinery at a single locus

*K. kingae* isolates contain either one or two copies of *rtxCrtxAtolC*[7,10,13]. To date, little is known about the genomics of the *K. negevensis* population due to both a lack of available isolates and few available whole genomes. To determine how many *rtx* loci are present in *K. negevensis*, we generated high-quality, complete genome sequences of 12 clinical isolates using hybrid assembly protocols (Supplementary Data 1). We performed de novo assembly of each sequenced strain, resulting in several completed circular genomes and plasmids. Assembled genomes were compared to each of the *K. kingae rtx* genes and complete *rtx* loci using BLAST (Fig. 2A). We found only a single copy of each gene in *K. negevensis*, with the genes appearing to be syntenic and highly conserved in all strains, suggesting that this island is found at a single site in the common ancestor of this species. These data suggest that the genomes of *K. kingae* and *K. negevensis* underwent unique structural changes since diverging.

To confirm this conclusion, pooled *K. negevensis* reads were then mapped to *K. kingae* strain KWG-1, which contains *rtx* genes at two loci: locus A (*rtxBrtxDrtxCrtxAtolC*) and locus B (*rtxCrtxAtolC*) (Fig. 2B–C). We hypothesized that, if the *K. negevensis rtx* system is organized like it is in *K. kingae* strains that contain a single copy of *rtxA*, then *K. negevensis* isolates with a single copy of *rtxA* would show no read-through from *rtxD* to *rtxA*, as these genes are non-syntenic in single *rtxA* copy *K. kingae* isolates. After mapping reads, we identified a large population of reads that span from *rtxD* to *rtxA* and no increase in read depth in the duplicated genes. These data suggest that there is a single copy of *rtxA* in the sequenced *K. negevensis* isolates and that the RTX machinery is encoded by a single locus.

### *K. kingae* RTX machinery genes are conserved across two loci in all isolates

In all *K. kingae* isolates, we found the *rtx* genes in two loci, named locus A and locus B (Fig. 2B–C). In some isolates, locus A contains the genes encoding the type-I secretion system (TISS; *rtxB* and *rtxD*) followed by *rtxC*, and locus B contains an additional copy of *rtxC*, followed by *rtxA* and *tolC* (Fig. 2B), an arrangement designated here as genotype I (Fig. 2B). Both copies of *rtxC* are preceded by a putative promoter for the *rtxCrtxAtolC* operon. Additionally, sequence analysis suggests that the duplicated genes are identical. Consistent with this finding, we confirmed expression of transcripts that contain primarily *rtxC* and *rtxA* in vitro by RT-PCR (Fig. 3A), strongly supporting operonic transcription of these genes, independent of the TISS. In other isolates, locus A contains an additional copy of *rtxA* and *tolC*, an arrangement designated genotype II and described previously in *K. kingae* strain KWG-1 (Fig. 2C)[10,18].

To determine the prevalence of the genotype I and genotype II arrangements, we designed a PCR-based approach that would specifically amplify the 3′ end of the additional copy of *tolC* found in locus A of genotype II isolates (Fig. 2B). A total of 124 clinical *K. kingae* isolates that were collected primarily in Israel between 1990 and 2012 were analyzed (Supplementary Data 1). Approximately 30% of these isolates contain genotype II *rtx* loci, suggesting that genotype II is prevalent in

the population. When strains were stratified by site of isolation, genotype II was strongly associated with strains recovered from patients with invasive disease ($X^2$ test of association; $p = 0.02$; df = 2) (Fig. 3B). A second screen was performed using an additional 131 global isolates collected between 1986 and 2015, revealing genotype II in approximately 35% of these isolates (Supplementary Data 3).

### Acquisition and reconstitution of fully functional RTX machinery

The *rtx* loci in *K. kingae* have features consistent with mobile genetic elements. Both loci are flanked by at least one putative transposon gene and repetitive sequences, and both have a G + C% that is lower than the G + C% in the rest of the genome (Figs. 2B–D and 3C, D). The putative transposon genes belong to the IS5 family of transposases (Fig. 2B, C).

To better understand the horizontal gene transfer event that resulted in the acquisition of *rtxA*, we performed a phylogenetic analysis of the proteins involved in toxin production and secretion, including RtxB (ATPase) and RtxD (membrane fusion protein) of the TISS, and the RtxA toxin. Due to difficulties in identifying RTX-associated acyl-carrier proteins in the database, RtxC was omitted from this analysis. NCBI RefSeq database was queried using the predicted amino acid sequences of each protein in *K. kingae* strain KWG-1. Homologous sequences with >50% sequence similarity were downloaded and used for phylogenetic reconstruction, allowing identification of RTX-associated proteins across 134 diverse bacterial species (Figs. S1–3). Interestingly, in all 3 trees there is a close relationship between *Moraxella* species genes and the *Kingella* toxin genes, signaling likely horizontal gene transfer given that these genera are found in different subdivisions of the proteobacteria (the γ-proteobacteria and β-proteobacteria, respectively).

The reconstructions of the RtxD and RtxB phylogenies are largely congruent and demonstrate that these *K. kingae* (and *Moraxella*) proteins have close relatives in other species within the *Neisseriaceae* family (Figs. S1–3). This is contrasted by the RtxA phylogeny, in which the *Moraxella*/*Kingella* clade is most closely related to *Pasteurellaceae* species, such as *Actinobacillus* spp. and *Mannheimia* spp. Of considerable interest, the *Kingella* RtxA is closely related to the ApxIIa toxin from *Actinobacillus pleuropneumoniae* (Fig S3). In addition to indicating a possible donor in the *Pasteurellaceae*, it is notable that *apxII* is found in a locus that does not contain homologs of *rtxB* and *rtxD*[19,20].

Phylogenetic incongruence between gene trees was measured by reducing our dataset to genomes where genes for the 3 proteins are found together. The RtxA phylogenetic matrix was tested against tree topologies for RtxB and RtxD. Statistical analyses show that the RtxA matrix strongly rejected the other topologies (RtxB tree: weighted KH test: $p = 0$, weighted SH test: $p = 0$, AU test: $p = 0.0064$; RtxD tree: weighted KH test: $p = 0$, weighted SH test: $p = 0$, AU test: $p = 0.00315$). Additionally, the reciprocal test forcing the RtxB alignment (weighted KH test: $p = 0$, weighted SH test: $p = 0$, AU test: $p = 1.83e-74$) or the RtxD alignment (weighted KH test: $p = 0$, weighted SH test: $p = 0$, AU test: $p = 0$) onto the RtxA topology strongly rejected congruence.

Perhaps most importantly, there are no close homologs to RtxA or RtxC in other species in the *Neisseriaceae*. Instead, the closest orthologous sequences are the Frp proteins in *N. meningitidis* and predicted Frp-related gene fragments in the non-pathogenic *Neisseria* species (Fig. S3). A full list of protein identities and similarities for proposed homologs in *K. kingae, K. oralis, K. denitrificans, K. negevensis, Moraxella bovis*, and *N. meningitidis* is shown in Supplementary Data 4. In addition, the G + C% content of the *rtxD* and *rtxB* genes in *K. kingae* (46% and 47%, respectively) is similar to the average G + C% of the *K. kingae* genome (46% for KWG-1) (Fig. 3C). The G + C% content of the *rtxC, rtxA*, and *tolC* genes at both loci in KWG-1 is considerably lower (Fig. 3C, D), consistent with the putative acquisition of these genes

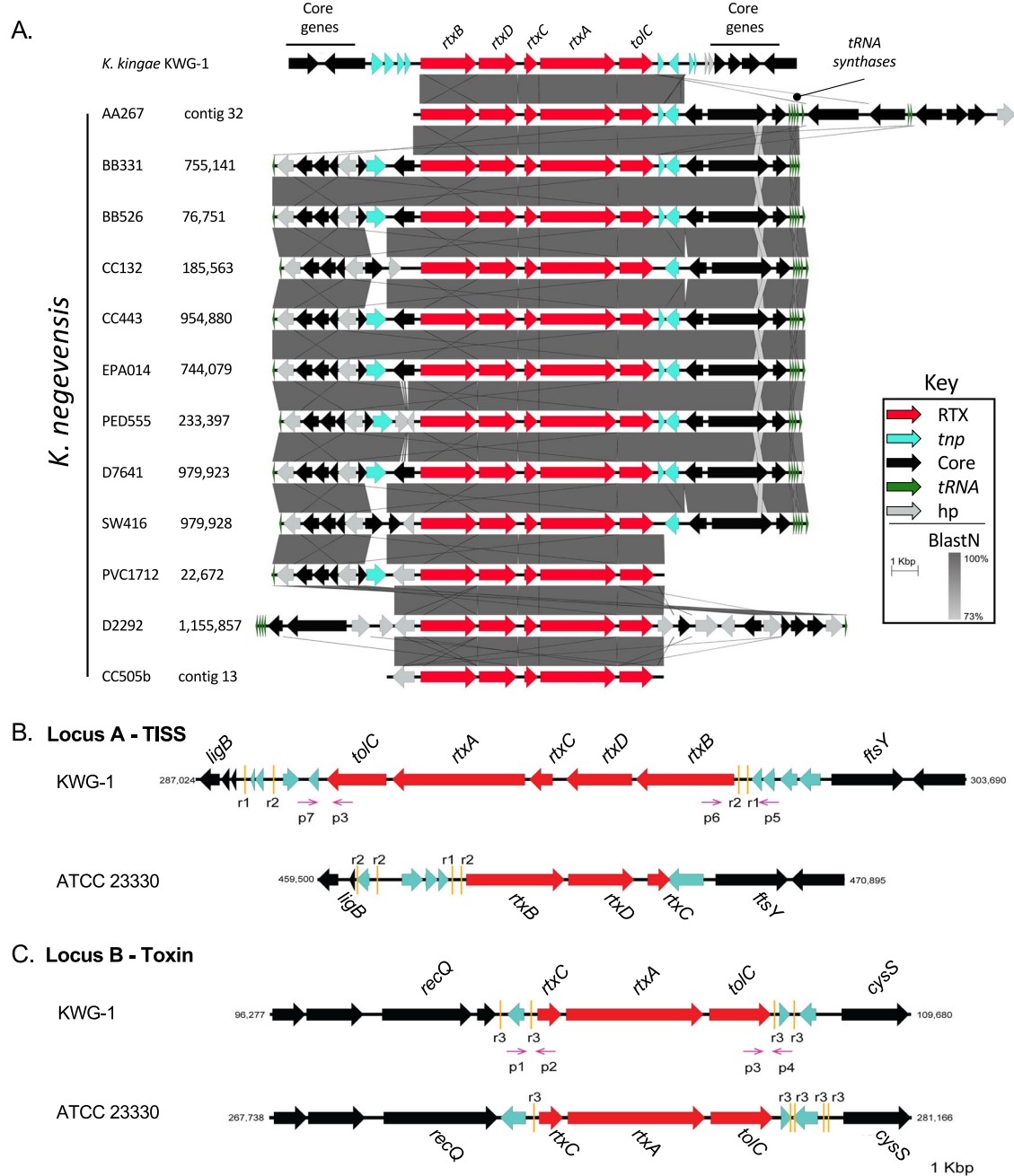

**Fig. 2 | The genome of *K. negevensis* encodes the RTX toxin machinery at a single locus. A** De novo whole-genome sequences were generated for each of the *K. negevensis* isolates used in this study. The RTX-associated genes were mapped in each strain. Region coordinates are shown for each assembly next to the strain name. These regions were compared to the RTX genes encoded by *K. kingae* KWG-1 by Blastn. Blastn identities are plotted in gray. RTX-associated genes are colored red, genes associated with DNA transposition (*tnp*) are shown in teal, core genes are shown in black, tRNA-synthase genes (*tRNA*) are shown in green, and hypothetical genes (*hp*) with no predicted functions are shown in gray. **B**, **C** Genomic maps of the *rtx* loci of representative *K. kingae* genotype I and II isolates (ATCC23330 and KWG-1, respectively). RTX-associated genes are shown in red. Each of these regions is flanked by several putative DNA transposition genes, shown in teal. Gold bars mark repeat sequences flanking each of these loci. Core genes are shown in black. Genotyping primers anneal at each end of these loci and are denoted by magenta arrows (p1–p7).

through horizontal gene transfer. Together these findings strongly suggest that the *rtxA, rtxC*, and *tolC* genes have an independent origin from *rtxB* and *rtxD* and that the full complement of genes was reconstituted as a functional toxin production system in the common ancestor of the pathogenic *Kingella* species.

### Whole-genome sequencing of *K. kingae* isolates

To investigate the role of the toxin duplication in *K. kingae* and to test for other factors associated with invasive disease, we sequenced 74 *K. kingae* and 12 *K. negevensis* isolates with Illumina HiSeq and combined these sequences with the genomes of 52 *K. kingae* strains in the NCBI Sequence Read Archive (SRA) (Supplementary Data 1). The average genome size for *K. kingae* is 2,003,405 bp (range: KK120, 1,943,140 bp; PYP8, 2,116,086). The average G + C% for the analyzed *K. kingae* strains is 46.75% (range: NCTC10746, 46.39%; KK242, 46.90%, assembly statistics data are listed in Supplementary Data 2).

As shown in Fig. 3E, a whole-genome phylogeny of *K. kingae* strains was constructed based on 125,599 nucleotide sites. For each

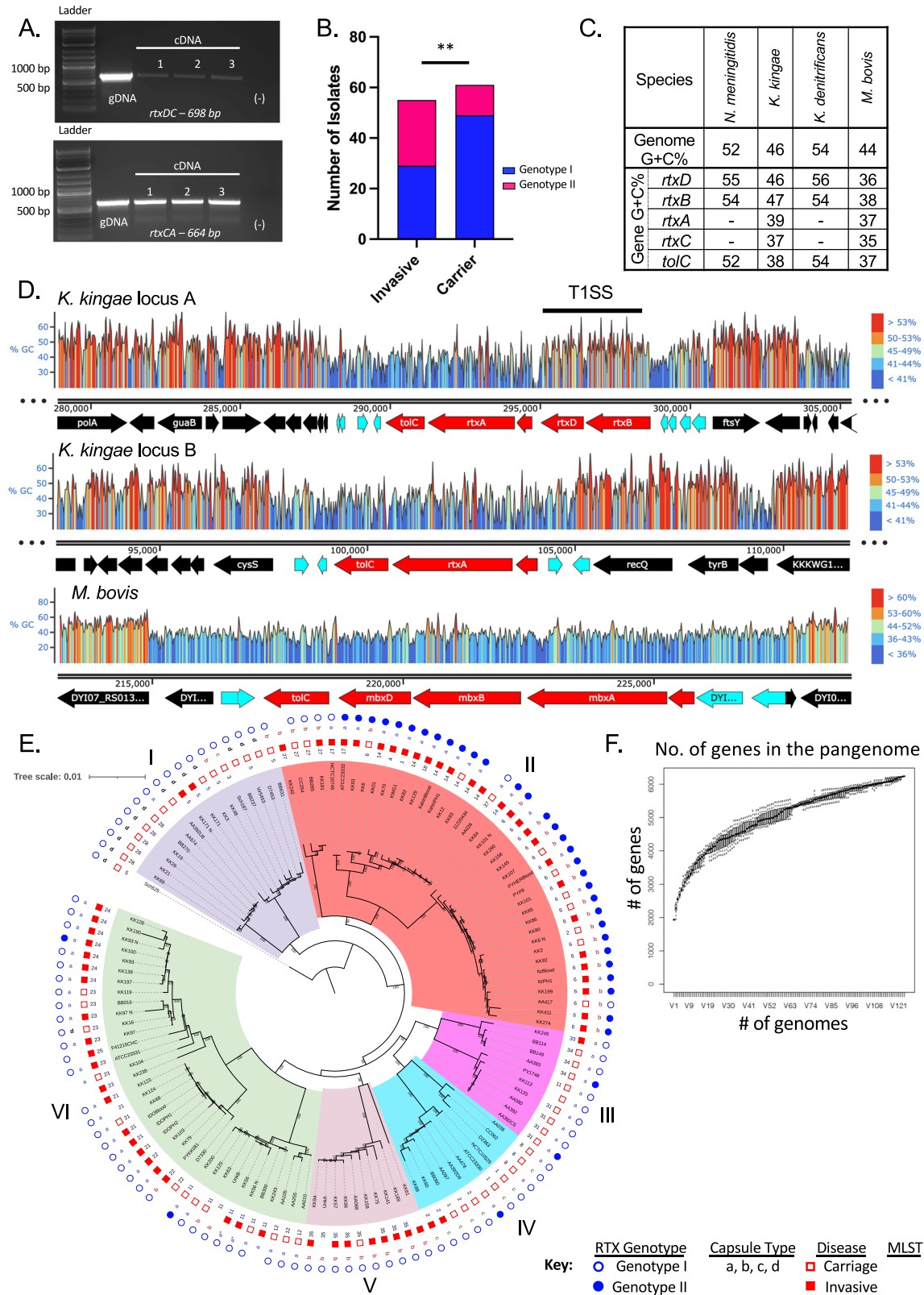

isolate, multilocus sequence type (MLST), polysaccharide capsule type, and clinical condition of the patient was mapped to the tree. The resulting tree can be divided into six clades (I–VI). The clades diverge along MLST and between capsule types. Clade I is comprised of primarily capsule type d carriage isolates. Clades IIa, VI, and VII contain primarily capsule type a isolates recovered from patients with invasive

*K. kingae* disease. Clades IIb and V contain primarily capsule type b isolates from patients with invasive disease. Clades III and IV contain primarily capsule types a and c carriage isolates. Isolates do not group by clinical site of isolation or year. Of the two clades that are enriched for invasive isolates, clade II is also enriched for *rtx* genotype II isolates, raising two possibilities: (1) the duplication of the *rtxA* gene is

**Fig. 3 | Reconstruction of the RTX machinery by *K. kingae*. A** To determine if these genes are transcribed as an operon in KWG-1, RNA was extracted, and RT-PCR was performed to identify read-through between the *rtxD* and *rtxC* genes (top) and between *rtxC* and *rtxA* (bottom). As a positive control, gDNA extracted from KWG-1 is shown. A representative gel of three, independent replicates is shown. **B** Presence of genotype I or II in a panel of 124 clinical isolates was determined using a PCR screening approach. Approximately 30% of the screened isolates are genotype II. Isolates were then stratified by isolation condition, invasive or carriage, and a $X^2$ test of association was performed. There is a significant association between *rtxA* copy number and disease state at the time of isolation ($p = 0.02$, df = 4). **C** Gene G + C% for each gene associated with the production of RTX family proteins from diverse bacterial species. G + C% for *rtxC* and *rtxA* are only shown for those species in which a close homolog could be identified. **D** Snapgene was used to generate a heatmap of G + C% across the toxin-associated genes in *K. kingae* strain KWG-1 locus A (top), KWG-1 locus B (middle), and the *Moraxella bovis mbx* pathogenicity island (bottom). Genome maps are labeled below the heatmap, with toxin genes colored red, DNA transposition genes colored teal, and core genes colored in black. TISS denotes the type-I section system genes. **E** A whole-genome phylogeny of *K. kingae*, rooted by *K. negevensis* Sch525, was constructed using publicly available genomes and an additional 88 genomes sequenced for this study. For isolates that appear in both the in-house and public databases, leaves marked with a "_N" denote those retrieved from NCBI. Each strain is marked with sequence type (ST), isolation condition (closed red squares denote strains from patients with invasive disease; open red squares denote carriage strains), capsular polysaccharide type, and *rtx* genotype (closed blue circles denote genotype II strains; open blue squares denote genotype I strains). The phylogeny has six distinct clades (named I–VI), which contain specific ST and polysaccharide capsule types. **F** A sampler's curve of unique genes identified in the pangenome of all *K. kingae* isolates included in this study is shown. The curve quickly saturates and begins to plateau, suggesting that this strain collection represents much of the genomic diversity of clinical *K. kingae* isolates. Box plots were generated by resampling genomes at random and recalculating the size of the pangenome within Roary. Boxes denote the 25th, 50th, and 75th quartiles and include the maximum and minimum values.

important for facilitating the pathogenesis of *K. kingae* disease, or (2) there is another genomic factor enriched in this clade that enhances invasion. This clade includes both global and historical isolates, limiting the possibility that each isolate in this group is from a single, local outbreak.

### Table 1 | Pangenome genes associated with Invasive and Carriage Isolates

|  | Gene | Odds_ratio | Annotation |
|---|---|---|---|
| Enriched in invasive | KKKWG1_0747 | 0.13919414 | Inovirus associated |
|  | KKKWG1_1631 | 0.16053512 | *hp* |
|  | KKKWG1_1448 | 0.17085714 | Peptidoglycan transpeptidase |
|  | KKKWG1_0749 | 0.19285309 | Inovirus associated |
|  | KKKWG1_0182 | 0.17769608 | Transpeptidase (NTF2-like) |
|  | ABJGGOKB_01117 | 0.13425254 | Pentapeptide repeat |
|  | KKKWG1_0242 | 0.15384615 | Ribonuclease VapC |
|  | KKKWG1_0241 | 0.15384615 | *hp* |
|  | KKKWG1_0029 | 0.19962453 | ABC Transporter |
|  | ABJGGOKB_00731 | 0.12882448 | Lytic Transglycosylase SLT domain |
|  | PIGKNEIA_01059 | 0.21125 | Inovirus associated |
| Enriched in carriage | ABJGGOKB_00216 | 12.88288288 | Transposase |
|  | ABJGGOKB_00217 | 12.88288288 | Endonuclease NucS |
|  | ABJGGOKB_00814 | 24.61363636 | Restriction Endonuclease |
|  | ABJGGOKB_01644 | 24.61363636 | Transposase |
|  | ABJGGOKB_01645 | 22.8 | Acyl-carrier protein |
|  | ABJGGOKB_01643 | inf | *hp* |
|  | ABJGGOKB_00501 | 13.02325581 | DNA binding cold-shock protein |
|  | ABJGGOKB_00500 | 13.02325581 | DMT family |
|  | ABJGGOKB_01824 | 13.02325581 | Helix-turn-helix transcriptional regulator |
|  | ABJGGOKB_01823 | 13.02325581 | Endopeptidase |
|  | ABJGGOKB_01822 | 13.02325581 | *hp* |
|  | KKKWG1_1450 | 9.837398374 | *hp* |
|  | ABJGGOKB_01646 | inf | *hp* |
|  | ABJGGOKB_00278 | 8.307692308 | *hp* |
|  | ABJGGOKB_00732 | 9.166666667 | Lytic Transglycosylase LysM |
|  | JIMKOJHD_01738 | 7.243902439 | GntP Family permease |

"*hp*" denotes hypothetical proteins with no predicted homologs in the BLAST database.

### Pangenome analysis reveals a limited pangenome and no other genes clearly associated with pathogenicity

To determine if there are additional genes that enhance virulence in isolates from invasive clades, we analyzed the pangenome using the Roary pipeline. A total of 6241 gene clusters were identified across these 124 isolates (Supplementary Data 5 and 6). Interestingly, though *K. kingae* is naturally competent, in a collectors' curve analysis, each new genome beyond 75 added very few new genes to the pangenome, suggesting that we have covered much of the pangenomic diversity of this species (Fig. 3F). The core genome is comprised of 1188 genes found in more than 95% of isolates (Supplementary Data 5). The pangenome is comprised of 5053 genes, including 3720 that are found in less than 15% of genomes. To determine which genes may be associated with invasive isolates, Scoary was used to perform a genome wide association study comparing invasive and carriage isolates, and comparing between clades (Table 1 and Supplementary Data 7). Eleven genes are specifically enriched in invasive isolates. As few of these genes have been characterized in *K. kingae*, functional predictions were made using BLASTp and HHPRED (Table 1), and none of these genes has a clear role in pathogenicity.

### *rtxA* copy number in isogenic strains influences toxicity in vitro but not virulence in animals

The RtxA toxin is required for virulence in an animal model of invasive disease using strain PYKK081, a genotype I isolate[10,12]. To examine whether duplication of *rtxA* influences virulence, we compared strains PYKK081, KKNB100 (PYKK081Δ*rtxA*), KK03 (genotype II), KK03ΔΔ*rtxA*, and KK03Δ*rtxA* (genocopies a genotype I). After a 1-h incubation at 37 °C with 5% $CO_2$, we observed high levels of RtxA in culture supernatants of PYKK081 and KK03. In contrast, KKNB100 and KK03ΔΔ*rtxA* secreted no RtxA. Secretion of RtxA was rescued by chromosomal complementation of *rtxCrtxAtolC* in KKNB100. KK03Δ*rtxA* secretes less toxin than KK03 (Fig. 4A). As shown in Fig. 4B, PYKK081 and KK03 are highly hemolytic; hemolysis was eliminated in KKNB100 and KK03Δ*rtxA* and was rescued by complementation of *rtxCrtxAtolC* in KKNB100. Hemolysis by KK03Δ*rtxA* was significantly reduced relative to KK03 (two-way ANOVA, $p = 0.03$). Cytotoxicity was assessed by LDH release from 16HBE-14o- human bronchial epithelial cells and was similar between KK03 and KK03Δ*rtxA* (Fig. 4C), likely due to the high sensitivity of these cells to this toxin in vitro.

To assess the impact of these genotypes on invasive disease, we employed the juvenile Sprague–Dawley model of disease. Five-day-old rats were inoculated intraperitoneally with $5 \times 10^7$ colony forming units of KK03, KK03Δ*rtxA*, or KK03ΔΔ*rtxA* and monitored for five days. With toxigenic strains, we observed significant death in the first 24 h post infection (h.p.i) (Fig. 4D). All animals infected with KK03 or KK03Δ*rtxA* succumbed to infection, while animals infected with KK03ΔΔ*rtxA* or

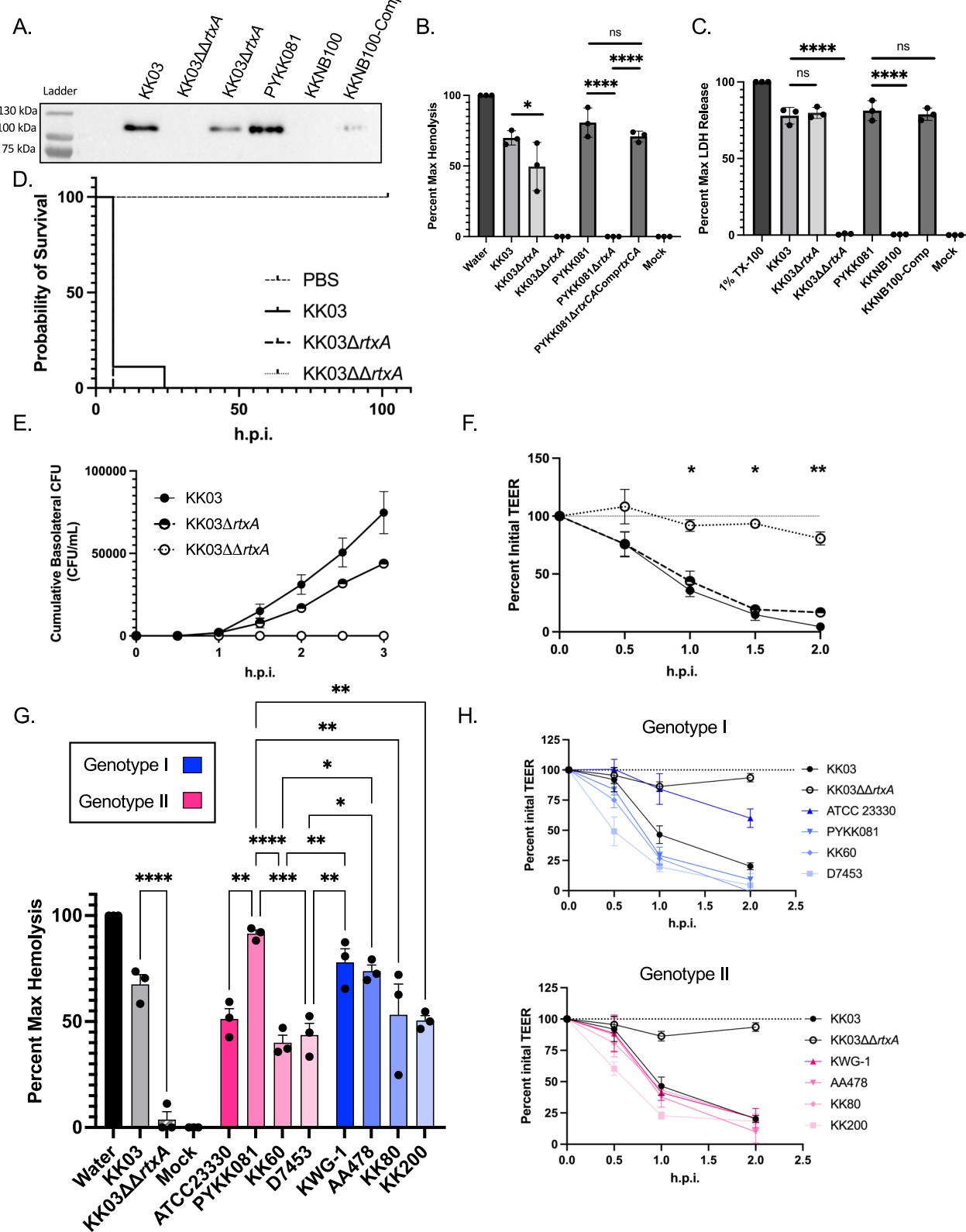

mock infected survived (Mantel-Cox test, both $p < 0.0001$). There was no difference in virulence between KK03$\Delta rtxA$ and KK03 (Mantel-Cox test).

To address whether RtxA influences bacterial invasion across the epithelial barrier, we adapted a model used for other respiratory pathogens to assess in vitro epithelial barrier integrity using 16HBE-14o-cells[21,22]. 16HBE-14o- cells were cultured at an air-liquid interface

(ALI) over the course of 10 days, allowing transepithelial resistance (TEER) to steadily increase. Cells were then transferred to serum-free media and infected apically with strain KK03 and the KK03 isogenic mutants and with strain PYKK081 and the PYKK081 isogenic mutant. As shown in Fig. 4E, F and Fig. S4A, B, RtxA-producing strains crossed the epithelial barrier, while strains KK03$\Delta\Delta rtxA$ and KKNB100 were attenuated. Strain KK03$\Delta rtxA$ transited the epithelial barrier slower

**Fig. 4 | In vitro and in vivo characterizations of genotype I and genotype II isolates. A** RtxA levels secreted into the culture supernatant were determined via western blot. The unprocessed blot is available in source data. To quantify differences in toxin activity, B. Liquid hemolysis of sheep red blood cells and C. LDH release from 16HBE14o- cells was examined using isogenic genotype I and genotype II strains. PYKK081, KKNB100, and KKNB100-comp are shown as controls. **B, C** Statistics were calculated with an ordinary one-way ANOVA with Sidak's multiple comparisons *$p = 0.0243$, ****$p < 0.0001$. **D** Kaplan–Meier survival curve of juvenile rat pups infected i.p. with KK03, KK03Δ*rtxA*, or KK03ΔΔ*rtxA*. PBS and KK03ΔΔrtxA are significantly different from WT ($p < 0.001$) and KK03Δ*rtxA* ($p < 0.001$). Statistics were calculated by the Mantel-Cox test, $N = 9$ for all groups. **E, F** 16HBE14o- cells were cultured at an ALI and infected at a multiplicity of infection of -10 on the apical surface in 1x MEM. Over the course of infection, cumulative transited CFUs in the basolateral chamber (**E**) were monitored every 30 min for 3 h post infection (h.p.i). Transepithelial resistance (**F**) was monitored

every 30 min for 2 h.p.i. KK03 is shown in filled circles, KK03Δ*rtxA* is shown in half filled circles, and KK03ΔΔ*rtxA* is shown in open circles. Statistical significance was calculated using a two-way ANOVA with Dunnett's multiple comparisons. **G** To quantify differences in RtxA efficacy, the same panel of clinical isolates was used in liquid hemolysis assays using sheep red blood cells. Genotype I isolates are shown in blue, and genotype II isolates are shown in magenta. Statistics calculated with a one-way ANOVA with Tukey's multiple comparison test. *$p < 0.05$, **$p < 0.01$, ***$p < 0.001$, ****$p < 0.0001$. **H** TEER was monitored over the course of infection of 16HBE-14o- semi-differentiated monolaters for genotype I (top) and genotype II isolates (bottom). Dashed line represents 100% initial TEER. For G-H, KK03 and KK03ΔΔ*rtxA* are shown as positive and negative controls, respectively. Genotype I isolates are colored in blue. Genotype II isolates are colored in magenta. For all experiments, points represent at least 3, independent biological replicates, and the averages are plotted with error bars representing the standard error of the mean (±1 SEM). Source data are provided in the attached source data file.

than KK03 (Fig. 4E). The strain background had a significant influence on the kinetics of this process, with PYKK081 crossing much faster and in higher numbers than KK03. The decrease in TEER correlated with transited CFUs (Fig. 4F and S4B). Both WT strains caused a rapid drop in TEER, indicative of significant disruption of tight junctions and a loss of epithelial barrier integrity. In contrast, strains KK03ΔΔ*rtxA* and KKNB100 caused no significant change in TEER over the course of infection (Fig. 4F and S4).

## Diverse genotype II isolates are not more pathogenic in vitro

To further investigate the impact of genotypes I and II on virulence, we selected eight clinical isolates from different clades across the *K. kingae* population structure (Fig. S4C). All of these isolates secreted similar levels of RtxA into the supernatant, suggesting that genotype II isolates do not generally produce more toxin than genotype I isolates (Fig. S4D). To address differences in activity after secretion, hemolysis and cytotoxicity assays were performed for each strain, revealing no consistent differences between genotype I and genotype II strains (Fig. 4G and S4E).

To examine the influence of genotype on the process of invasion, we used the model with polarized 16HBE-14o- cells. Cultures were infected apically at multiplicity of infection of -10. As shown in Fig. 4H and Fig. S4F–I, over the course of infection, both genotype I and genotype II isolates were able to efficiently disrupt TEER with similar kinetics, resulting in similar bacterial transit (two-way ANOVA, Fig. S4F–I).

## Discussion

*K. kingae* is an oropharyngeal commensal that is most common in young children and is increasingly recognized as a pathogen capable of causing severe systemic disease[1,2]. Recent studies have identified five major virulence determinants that are conserved across this species: type IV pili, a trimeric autotransporter adhesin called Knh, a polysaccharide capsule, an exopolysaccharide, and a broadly active RTX cytotoxin called RtxA[7]. The lipopolysaccharide-associated exopolysaccharide and the RtxA toxin are found only in pathogenic *Kingella* species[11,23]. In this study, we reconstructed the evolutionary history of *rtxA* in the pathogenic *Kingella* species based on a large collection of isolates from both invasive disease and oropharyngeal carriage.

In *K. negevensis*, the genes encoding the toxin-associated machinery (*rtxBrtxDrtxCrtxAtolC*) are found in a single locus. In contrast, using a novel PCR-based approach on a collection of 255 clinical isolates, we determined that all *K. kingae* strains contain *rtx* genes at two loci, and 30-35% have a duplication resulting in a second copy of *rtxA* and *tolC* downstream of the TISS genes and the *rtxC* gene (Fig. 2B, C). These results are reminiscent of *Actinobacillus pleuropneumoniae*, a porcine-specific, respiratory pathogen that contains genes for multiple related RTX toxins in the Apx family. Of the Apx toxins, ApxI is strongly hemolytic, ApxII is weakly hemolytic,

ApxIII is non-hemolytic, and ApxIV is a "clip and link" protein[24–26]. All *A. pleuropneumoniae* isolates produce ApxIV, in combination with one or two other Apx toxins, depending on the respective serotype of the strain. Interestingly, the closest *apx* homolog of *Kingella* RtxA is found in a locus that lacks *rtxB* and *rtxD* homologs[24–26]. Changes in toxin functionality and regulation across different loci could be prevalent in *K. kingae* and may account for the association between *rtxA* duplication and invasive disease.

Our analysis suggests that the *rtxCrtxAtolC* locus was acquired horizontally in the common ancestor of *K. kingae* and *K. negevensis*. While it is unclear from where *K. kingae* acquired this island, the island is closely related to the *mbx* genes encoded by *Moraxella* species from sheep, goats, and cattle. *Moraxella* species inhabit a similar niche to *Kingella* species in the upper respiratory tract, and *M. bovis* employs an RTX toxin as a major virulence factor in ocular disease[27,28]. Toxin-associated genes present in *M. bovis* were also likely acquired as part of a genomic islet in a single event[28]. Based on the absence of close *rtxA* genes in other *Neisseriaceae*, the presence of *rtxB* and *rtxD* homologs in other *Neisseriaceae* including close relatives such as *Alysiella crassa* that do not have an *rtxA* homolog, as well strong statistical phylogenetic incongruence between RtxB/RtxD and RtxA phylogenies, our analysis suggests an independent acquisition of the *rtxCrtxAtolC* locus in the common ancestor of pathogenic *Kingella* species. This is bolstered by the finding of closely related toxins (ApxIIa) from *Actinobacillus pleuropneumoniae*, which do not encode a TISS in the same locus as the toxin genes[19,20]. If acquired in isolation, this locus would have resulted in a nonfunctional toxin production system in the pathogenic *Kingella* species because it lacked a TISS for export. However, our analyses suggest that the presence of a preexisting TISS (*rtxBrtxD*) likely allowed reconstitution of a complete toxin production and secretion system through the process of co-option. Given the putative reconstitution of the toxin system in the common ancestor of pathogenic *Kingella* species and the close relationship of the entire locus with *Moraxella* species, it seems likely that this ancestor may also have been the donor to *Moraxella* species.

The preexisting TISS is most closely related to genes of the *frp* operon, which encodes the distantly related RTX family protein, FrpC, and is intact in *N. meningitidis* but not in *N. gonorrhoeae* or commensal *Neisseria* speices[29,30]. Though the specific function of FrpC is unclear, it is thought to act as an additional adhesin that is dispensable for virulence in a mouse model[31–35]. This putative genetic reconstitution event provides an important example of gene co-option as a major contributing factor in the evolution of bacterial virulence in a commensal genus. Previous work has described co-option of phage-associated proteins for novel functions, including the *Caulobacterales spmX* gene, the type IV and type VI secretion systems, and a bacteriocin secreted by *Pseudomonas syringae*[36–39]. However, the co-option of the components required for toxin elaboration is unique to *K. kingae*. Given our virulence model and genomic data, we suggest that reconstitution of a

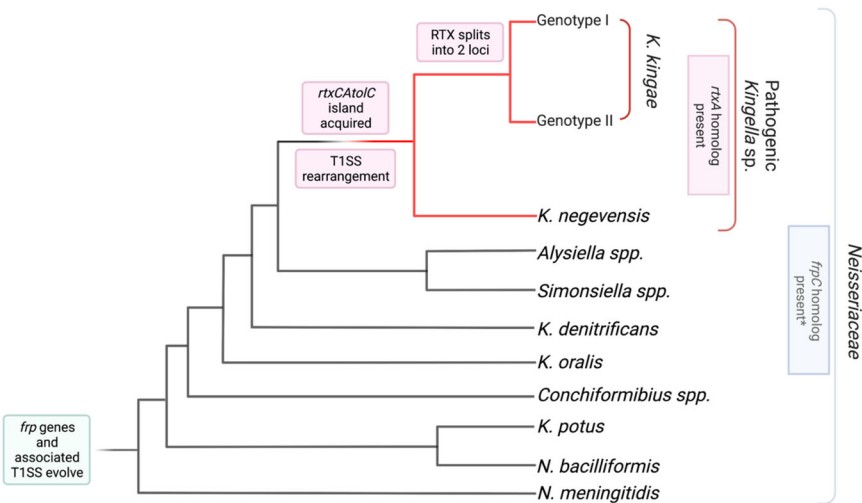

**Fig. 5 | Model of evolution of the *rtx* toxin and the associated T1SS in *K. kingae*.** The Frp-associated genes were acquired by the last common ancestor of *N. meningitidis* and *Kingella species* and have been maintained in descendants through the divergence of *N. meningitidis* and the *Kingella* genus. Early branching species of *Kingella* are not associated with invasive disease, though they maintain the Frp machinery. Homologs of the *rtxC* and *rtxA* genes were then acquired by the last common ancestor of *K. negevensis* and *K. kingae*, without an accompanying TISS.

The TISS associated with the *frp* genes was co-opted by the *rtxCrtxAtolC* system to both facilitate the development of the pathogenic *Kingella* species and confer a fitness advantage. As *K. kingae* and *K. negevensis* diverge, the RTX machinery underwent two recombination events in *K. kingae*, which first split the genes into two loci in the genome, then resulted in the duplication of *rtxC* and *rtxA* in a specific, highly invasive lineage.

complete RTX system by this mechanism is an essential and defining feature in the evolution of invasive disease by this genus.

We sought to test the association between *rtxA* copy-number variability and invasive disease in vitro and in vivo. Using a panel comprised of four one-copy and four two-copy isolates, we assessed secreted RtxA levels using Western blotting. We then used in vitro assays of toxin function, including hemolysis, cytotoxicity, and epithelial barrier breach. Using genotype I and II strains and isogenic derivatives, we observed a reduction in toxin secretion and hemolysis when one-copy of the *rtxA* toxin was disrupted. Using in vitro assays of cytotoxicity, we could not differentiate between diverse genotype I and II isolates in a panel of clinical isolates, suggesting that our in vitro models may be influenced by other qualities of clinical strains that cloud toxin-specific effects. Infection is a multifactorial process, and to date studies focused on the pathogenesis of *K. kingae* disease have taken a reductionist approach to study the role of one virulence factor in isolation. By leveraging the clinical panel described here, future studies can begin to untangle the interactions between RtxA and other factors.

Moreover, while our whole-genome sequencing suggests that the *rtxA* genes are identical between the two *rtx* loci, the promoters of the proposed *rtxCrtxAtolC* operons appear to be unique to each copy, suggesting that there may be meaningful differences in regulation in two-copy isolates under different environmental conditions.

We propose the following model of the acquisition, co-option, and duplication of the *rtx* toxin genes and the associated TISS in *Kingella*. A common ancestor of all genera in the *Neisseriaceae* contained the Frp system and the necessary TISS required for the secretion of the FrpC RTX protein (Fig. 5). Following acquisition, the functionality of the Frp system was maintained only in the lineage that later evolved into *N. meningitidis*. While the TISS remained largely intact, other genera accumulated mutations in FrpC, ablating its activity and resulting in the *frp* fragments that we observe in contemporary bacterial isolates. As such, we identified the components of a homologous TISS in several members of the *Kingella* clade within the *Neisseriaceae*.

The common ancestor of the pathogenic *Kingella* species acquired *rtxCrtxAtolC*. Following acquisition of these genes, we

propose that toxin secretion occurred via the existing TISS, which was previously associated with the Frp system. This acquisition substantially increased the fitness of this ancestral species and resulted in the appearance of invasive disease by *Kingella* species. Thus *K. kingae* co-opted the FrpC toxin-associated secretion genes, now *rtxBrtxD*, to export RtxA rather than FrpC. Subsequent recombination events brought the TISS into close genetic proximity with the *rtxC* gene, but the *rtxDrtxB* operon remained transcriptionally independent from *rtxCrtxAtolC*, as we observe in *K. kingae*. In *K. kingae*, further reorganization of the genome split the toxin machinery across two distant loci, duplicating the *rtxC* gene. Finally, a clonal group of *K. kingae* associated with invasive disease underwent another round of genomic reorganization, resulting in a second duplication event establishing two copies of *rtxCrtxAtolC* in the genome.

While it is possible that the reconstitution of the RTX machinery occurred in a common ancestor of all *Kingella*, we find it unlikely that the commensal species lost the toxin genes. Additionally, it is possible that *rtxCrtxAtolC* was acquired along with genes for a functional TISS but these genes were subsequently lost, being functionally replaced by the TISS associated with the Frp system. It is interesting to speculate that the secretion system encoded by the *rtxBrtxD* genes may have been conserved over time due to the secretion of other proteins from the bacterial cell, though further work is required to characterize the secretome of *K. kingae* and identify those proteins specifically secreted by the *rtxBrtxD*-encoded TISS.

## Methods
### In silico analysis of diverse *Kingella* isolates
**Whole-genome sequencing.** Isolates used in this study and all available metadata are listed in Supplementary Data 1. Clinical isolates for sequencing (88 isolates, primarily recovered from patients in Israel) and genotyping (an additional 131 global isolates) were selected from a large collection of more than 500 *K. kingae* isolates collected from the early 1990's[40,41]. Additionally, the sequencing reads from 52 publicly available genomes were downloaded from the NCBI Sequence Read Archive (SRA) database on August 5, 2020. Publicly available genomes include isolates from as early as 1966. Genomic DNA (gDNA) for sequencing and genotyping was extracted using the Wizard Genomic

DNA purification kit (Promega) as directed for Gram-negative bacterial species. The gDNA concentration was determined via NanoDrop 2000 (Thermo Scientific).

Extracted gDNA was fragmented and barcoded using the Nextera XT kit (Illumina), following the manufacturer's instructions. Genomes were sequenced on an Illumina HiSeq 2500 machine using the 100SR single end read protocol. Sequencing adapters and low-quality ends with a PHRED score less than 15 were trimmed from reads using Trim Galore v. 0.6.4[42]. Trimmed reads and reads downloaded from the SRA databases were assembled using SPAdes assembler v. 3.14.0 and default settings for single end Illumina reads[43,44]. Quality control data for both trimming and assembly are shown in Supplementary Data 2 and was calculated using QUAST. Assembled scaffolds were annotated using Prokka v. 1.14.6, using the following flags and options: --addgenes --kingdom Bacteria --genus *Kingella* --species *kingae*[45]. Finally, nucleotide and amino acid FASTA files generated as an output from Prokka for all isolates were compiled into two databases for local blast searches. All reads and assemblies are available through SRA (BioProject PRJNA896475). To validate that all strains were members of the species *K. kingae*, the average nucleotide identity score (ANI) was calculated for each isolate relative to the isolate *K. kingae* strain KWG-1 with FastANI v.1.32[46]. Additionally, FastANI was used to generate an output matrix later plotted as a heatmap in R v.4.1.2. Quast v. 5.0.2 was used to determine the quality of all assemblies (data summarized in Supplementary Data 1 and Supplementary Data 2), using *K. kingae* KWG-1 as the reference sequence[47]. Sequence type was assigned using the *K. kingae* PasteurMLST scheme[15,47,48].

## Whole-genome sequencing of *K. negevensis* strains

*K. negevensis* gDNA was extracted using the Wizard HMW DNA extraction Kit (Promega) following the manufacturer's instructions. Following extraction, DNA quality was determined using TapeStation Genomic DNA Screentape (Agilent) and quantified using a NanoDrop 2000 (Thermo Scientific). For sequencing, libraries were generated using the Rapid Barcoding Sequencing Kit (SQK-RBK004, Oxford Nanopore Technologies), using 400 ng of template DNA, and following the manufacturer's instructions. Sequencing was performed on the MinION Mk-1B using the Spot-ON Flow Cell R9 (Lot 11002153, Oxford Nanopore Technologies). Base calling was performed using the MinKNOW software with increased stringency. Low-quality reads and reads less than 1 kB in length were removed with FiltLong. Short read sequencing was performed by SeqCenter (Pittsburgh, PA). Sample libraries were prepared with the Illumina DNA prep kit and IDT 10 bp UDI indices. Sequencing was performed on an Illumina NovaSeq600. Demultiplexing, preliminary quality control, and adapter trimming was performed with bcl-convert v4.0.3. De novo assemblies of each isolate were performed with the Unicycler genome assembly tool[49]. Genomes were annotated using Prokka, as above, and the RTX-associated genes were mapped with Blast and visualized using Easyfig[50].

## Pangenome characterization

The GFF output file from Prokka was analyzed with Roary v.3.13.0 to determine genes in the core and pangenome[51]. Roary was run with the following options selected: create multiFASTA alignments of core genes using PRANK (−e), generate a fast core gene alignment with MAFFT[52] (−n), at least 95% percent of isolates must encode a gene for the gene to be considered a core gene (-cd 95), generate R plots (-r), and cluster paralogs (−s). All other settings were default. To identify known virulence factors in the core and pangenome, assembled contigs were analyzed against the VFDB and MEGARES database with Abricate v.1.0.1 and NCBI AMRFinder v.3.9.3, using default settings[53–56]. Scoary v.1.6.16 was used to perform genome wide association studies to identify coding sequences associated with invasive isolates, relative to carrier isolates, with default settings[57].

## Phylogenetic reconstructions

Single nucleotide polymorphisms (SNPs) were identified in trimmed sequencing reads from all genomes that passed quality control cutoffs with Snippy v. 4.6.0 by generating a whole-genome alignment against *K. kingae* KWG-1 as a reference (Genbank file downloaded from NCBI September 01, 2020)[58]. The output alignment file was cleaned using Snippy clean module, and a phylogenetic tree was reconstructed in RAxML v.8.2.4, along with 100 bootstraps, the GTR substitution model, GAMMA model of rate heterogeneity, rapid hill climbing (-f d), and a random number seed of 1977[59]. These settings were used for all nucleotide trees, unless otherwise noted. The impact of recombination in the phylogenetic trees was assessed using ClonalFrameML v.1.12, with default settings[60]. Trees were visualized and annotated with metadata in FigTree v.1.4.4 and/or iTOL[61].

Amino acid sequences of RtxA, RtxB, or RtxD from *K. kingae* KWG-1 were used to query protein databases on NCBI with BlastP. Representative sequences of the closest homologs from Gram-negative bacteria were downloaded and aligned using ClustalW v.2.1 (RtxA: 1013 parsimony-informative sites; RtxB: 548 parsimony-informative sites; RtxD: 494 parsimony-informative sites)[62]. Tree reconstructions were performed in IQTree v.2.2.0[63]. The optimal substitution model was selected with Modelfinder[64]. The LG + R10 substitution model was selected for RtxB, the LG + F + R7 substitution model was selected for RtxD, and the Blosum62+F + R5 substitution model was selected for RtxA. Branch support was calculated with 1000 ultrafast bootstraps. Trees were visualized using iTOL.

Statistical tests on phylogenetic trees were performed in IQTree v.2.1. Using the phylogenies constructed above, a set of 134 species were selected to include diverse bacterial species. For each species, one representative sequence for each gene was collected using BlastP. Gene sequences were aligned again with ClustalW v.2.1, and phylogenies were reconstructed using the same models as above. To test for incongruence between trees, IQTree was used to search the tree space for the RtxA alignment constrained by the phylogenetic tree topology for RtxA, RtxB, or RtxD. For each, the p-values of weighted Kishino-Hasegawa test (KH), weighted Shimodaira-Hasegawa test (SH), and the approximately unbiased (AU) test were calculated[65–67].

## Genotyping clinical isolates

Primers used in this study are listed in Table 2. *K. kingae* strain KK01 (a non-spreading, non-corroding stable derivative of clinical isolate 269-492) was used to design primers against the four ends of the *rtx* loci and the previously described capsule typing primers[41]. PCR assays were performed on gDNA extracted as described above. For the *rtx* loci, the presence of an amplicon of the expected size on an agarose gel was used to determine presence or absence. The multiplex capsule typing assay produces amplicons of different size for each of the four capsule types[41]. A gene deletion mutant was used as a negative control to demonstrate PCR specificity. Additional strains not included in sequencing but genotyped by PCR are listed in Supplementary Data 3. To determine the association between these genotypes and disease state, a Fisher's exact test was performed in Graphpad Prism v.9.2.0.

## In vitro characterization of clinical isolates

*K. kingae* isolates used in genotyping PCR assays were grown on chocolate agar at 37 °C with 5% $CO_2$, as previously described[9]. *K. negevensis* strains were grown on brain heart infusion (BHI) agar supplemented with 10% sheep blood for 24 h at 37 °C, 5% $CO_2$[18].

Mutants were generated as described previously[9,12,68]. Briefly, NEB Builder was used to design primers for three fragment Gibson assembly cloning (Table 2). Plasmids were constructed in the pUC19 backbone using NEBuilder Hi-Fi DNA Assembly Mastermix (New England Biolabs) following the manufacturer's instructions. Vectors were linearized and subsequently introduced into *K. kingae* by natural

**Table 2 | Primers used in this study**

| | Primer name | Sequence (5′ → 3′) | Product (bp) | Reference |
|---|---|---|---|---|
| *RTX typing*[a] | | | | |
| p1 | rtxtrunc_up_fwd | TGCATCATGTGCATGGTTTG | ~1100 bp | This study. |
| p2 | rtxC_trunc_rev | ACATACTGTTGCGTTTCAATA | | This study. |
| p3 | tolc_trunc_fwd | AATTAAACTAGCAGTTCGCCAAG | ~1600 bp | This study. |
| p4 | rtxtrunc_down_rev | AAGCAGCCTGCACATTATTC | | This study. |
| p5 | rtxfull_up_fwd | TCATGCAGCCAATATTCATGAC | ~500 bp | This study. |
| p6 | rtxB_rev | TGATAGTGCGCAAGGATAATTAAC | | This study. |
| p7 | rtxfull_down_rev | TGAGGTATAAACGCTTGCAGTG | ~2200 bp | This study. |
| *Capsule typing* | | | | |
| 1 | csamultiF | AGTACAGAACACTTGTTGTTGC | ~2000 bp | 41 |
| 2 | csamultiR | AACATTGGCGCAGACAAATTC | | 41 |
| 3 | *csb*multiF | AGATTGGTGGACTTTATATGGTAATTATG | ~1500 bp | 41 |
| 4 | *csb*multiR | AAATAGAATATTGCGACTGTGCG | | 41 |
| 5 | cscmultiF | CATTAGCATTGATGCCATTTATGAAC | ~1000 bp | 41 |
| 6 | cscmultiR | CGATTGATGACTATTAAACCTTCGG | | 41 |
| 7 | *csd*multiF | AAAGGTAAATATCAATTTGCAATTATTTGC | ~500 bp | 41 |
| 8 | *csd*multiR | CTTAATAGGACATCATCACCCAAATC | | 41 |
| *qPCR* | | | | |
| 1 | rtxAC_operon_fwd | TCGTTCAGACTGCCTCCTTT | ~700 bp | This study. |
| 2 | rtxAC_operon_rev | GTTGGTTCGAGATGGGATGC | | This study. |
| 3 | prtxC_Full_fwd | AAAAGCATCAGCCACGATG | ~700 bp | This study. |
| 4 | prtxC_rev | AATTCAACCTTGCCCAACTG | | This study. |
| *Cloning* | | | | |
| pCE81*rtxCAtolC* | 81tolCcomp_rev_KpnI | ACGTGGTACCCGCTGTCAATTGTGCATATTTTATATG | | This study |
| | rtxCA_trunc_comp_F | AGCTGTCGACTCATCAGCTTTGGAATGACAAC | | This study |
| pUCΔ*rtxA*1144 | rtxAdelupTet_fwd | ttgtaaaacgacggccagtgCAAGGTCAAGATGCTGTG | | This study |
| | rtxAdelupTet_rev | tttcctccatAATTCACCTCAATTTTCAAATTTG | | This study |
| | TetORF_fwd | gaggtgaattATGGAGGAAAATCACATG | | This study |
| | TetORF_rev | atcgcattatCTAAGTTATTTTATTGAACATATATCGTAC | | This study |
| | rtxAdeldownTet_fwd | aataacttagATAATGCGATAGCACCATTTG | | This study |
| | rtxAdeldownTet_rev | atccccgggtaccgagctcgTGGTAATGGGTAGCATTTC | | This study |
| pUCΔ*rtxA*1228 | RtxFullup_3_fwd | cgacgttgtaaaacgacggccagtgTTGGATTACTTGTTGAGTC | | This study |
| | RtxFullup_3_rev | ttgacagtttatGTTGTTGCTAAGTTCATAATTC | | This study |
| | TetM_fwd | acttagcaacaacATAAACTGTCAATTTGATAGCG | | This study |
| | TetM_rev | ctttcaagaataCCATATTTATATAACAACATAAAATACAC | | This study |
| | RtxFulldown_fwd | tatataaatatggTATTCTTGAAAGAATGCGAATTTC | | This study |
| | RtxFulldown_rev | tacgccaagcttgcatgcctgcaggACATAATACGTCTTAGTAATCAAAAC | | This study |

[a]Approximate annealing locations of the genotyping primers are indicated on Fig. 2B, C.

transformation followed by selection on chocolate agar plates supplemented with the appropriate antibiotic (Table 3). Mutations were confirmed by preparing gDNA, amplification by PCR, and Sanger sequencing.

To complement *rtxA* in PYKK081, the primers "81tolCcomp_rev_KpnI" and "rtxCA_trunc_comp_F" were used to amplify *rtxCrtxAtolC*, and the resulting amplicon was digested with KpnI/SalI and ligated into KpnI/SalI-digested pCompErm, generating pCE-81*rtxCAtolC* (Tables 2–3). The complementation plasmid was linearized with SfoI and transformed into KKNB100 via natural transformation. To generate KK03ΔΔ*rtxA* and KK03Δ*rtxA*, *rtxA*$_{LocusA}$ was replaced with the *tetM* tetracycline resistance cassette using a four-fragment assembly and the NEB Hifi Assembly Kit. The resulting plasmid was linearized and transformed into an existing *rtxA* transposon mutant (60H11T1) to generate KK03ΔΔ*rtxA* or into strain KK03 to generate KK03Δ*rtxA*.

## RNA extraction

Bacteria were grown for 20 h at 37 °C with 5% $CO_2$, resuspended in 3 mL brain heart infusion (BHI) broth to an $OD_{600}$ of 0.8, and incubated at 37 °C with 5% $CO_2$ for 1 h. A volume of 1 mL was removed and centrifuged for 15 min at 4000 x $g$ to pellet bacteria. Pellets were resuspended in 1 mL of TRIreagent and incubated for 10 min at room temperature (RT). RNA was extracted from the aqueous phase by the addition of 200 μL of chloroform and centrifugation at 10,000 x $g$ for 10 min. The aqueous phase was removed and added to 600 μL of 100% isopropyl alcohol. Glycogen (ThermoFisher) was added to each sample to aid in pellet visualization, and samples were incubated for 10 min at RT. Nucleic acids were pelleted by centrifugation at 10,000 x $g$ for 30 min. The pellets were washed with 1 mL of 70% ethanol prepared with DEPC-treated water and were then treated with DNAse I to remove contaminating DNA. DNase-treated RNA was then re-extracted as described above. cDNA was generated using qScript cDNA Supermix

## Table 3 | Strains and constructs used in this study

| Name | Description | Reference |
|------|-------------|-----------|
| *Strains* | | |
| KK03 | Spreading/corroding derivative of isolate 269–492 | 9 |
| 60H11T1 | KK03 with 1 *rtxA* disrupted by *aphA3-Tn* | 9 |
| KK03Δ*rtxA* | KK03 with the locus A *rtxAtolC* genes replaced with *tetM* | This study |
| KK03ΔΔ*rtxA* | 60H11T1 with the remaining copy of *rtxA* replaced with *tetM* | This study |
| PYKK081 | Septic Arthritis isolate | 70 |
| KKNB100 | PYKK081Δ*rtxA::kan* | 12 |
| KKNB100-Comp | Chromosomal complement of KKNB100 with *rtxCAtolC* | This study |
| *Plasmids* | | |
| pCompErm | Chromosomal complementation plasmid | 71 |
| pCE81*rtxCAtolC* | pCompErm encoding the *rtxCrtxAtolC* operon | This study |
| pUC19 | pUC high copy-number cloning vector | 72 |
| pUCΔ*rtxA*1144 | pUC19 with *tetM* flanked by surrounding KK03 5' and 3' regions of locus A | This study |
| pUCΔ*rtxA*1228 | pUC19 with *tetM* flanked by 5' and 3' regions of locus B in KK03 for targeting locus A | This study |
| pHSX*tetM*4 | Source of the *tetM* tetracycline resistance cassette | 73 |

(Quanta Bioscience) and 1 µg of RNA. Quantitative PCR (qPCR) was performed using primers listed in Table 2.

### Polyclonal antiserum generation
Purified recombinant RtxA (a gift from Nataliya Balashova) was sent to Cocalico Biologicals, Inc. (Stevens, PA) for polyclonal antiserum generation in a guinea pig according to their standard antibody production protocol. The resulting RtxA-specific polyclonal antiserum is designated GP-23. GP-23 was validated by western blot and ELISA against purified, recombinant RtxA, as well as against RtxA-deficient mutant strains.

### Western blotting
*K. kingae* strains were resuspended into heart infusion (HI) broth to an $OD_{600}$ of 0.8 and incubated at 37 °C with 5% $CO_2$ for 1 h. A 1 mL aliquot was removed and centrifuged at 14,000 x *g* for 2 min. To the clarified supernatant, trichloroacetic acid (TCA) was added to final concentration of 10%. After 15 min on ice, the precipitated proteins were collected by centrifugation at 21,000 x *g* for 10 min at 4 °C and were washed with 1 mL ice-cold acetone. After centrifugation at 21,000 x *g*, the resulting pellet was resuspended in 1x sodium dodecyl-sulfate polyacrylamide gel electrophoresis (SDS-PAGE) loading buffer and boiled for 5 min, and an aliquot was separated using 10% SDS-PAGE. After transfer to nitrocellulose, the blot was blocked in 5% skim milk in 1x PBS. The blot was then probed with a 1:2000 dilution of GP-23 for 1 h at ambient temperature, washed with Tris-buffered saline supplemented with 0.1% Tween-20, probed with a 1:10,000 dilution of an anti-guinea pig-horseradish peroxidase-conjugated secondary antibody (Sigma; product no. A7289) for 30 min at ambient temperature, washed again, and developed using a Syngene G-Box XT4 system (Frederick, MD) after exposure to a chemiluminescent substrate (ThermoFisher Supersignal Plus).

### Hemolysis
Hemolysis was assessed with bacteria grown for 20 h at 37 °C with 5% $CO_2$ and resuspended in 3 mL of HI broth to on $OD_{600}$ ~ 0.8. Suspensions were then diluted 1:100 in HI and mixed in equal volumes with 1% washed sheep red blood cells in 1x PBS supplemented with 0.492 mM

$CaCl_2$ and 0.900 mM $MgCl_2$. After incubation for 30 min at 37 °C with 5% $CO_2$, each reaction was pelleted at 1000 x *g* for 2 min. The absorbance at 415 nm was measured for each supernatant. Cultures were standardized to a maximum hemolysis condition generated by adding water in place of bacteria. Percent maximum hemolysis was calculated for each strain relative to a water lysis control. Statistical analysis was performed in Graphpad Prism v.9.2.0.

### Cytotoxicity
Cytotoxicity was assessed by LDH release from 16HBE-14o- cells. Cells were seeded in 1x MEM supplemented with 10% FBS (Corning) at a density of $2.5 \times 10^4$ in a 96-well tissue culture plates (Greiner) in triplicate and grown at 37 °C with 5% $CO_2$ for 48 h. Bacteria were grown for 20 h at 37 °C with 5% $CO_2$ and resuspended in 3 mL of HI broth to on $OD_{600}$ ~ 0.8. After resuspension, the culture media for the 16HBE-14O-cells was replaced with 100 µL of prewarmed, serum-free 1x MEM. A 5 µL aliquot of the resuspended culture was added (M.O.I. ~100), and the plates were centrifuged for 2 min at 1000 x *g* followed by incubation for 1 h at 37 °C with 5% $CO_2$. The infected cells were once again centrifuged for 2 min at 1000 x *g*, and 50 µL of supernatant was removed for quantification. LDH in the cell supernatants was quantified using the Cytotoxicity Detection kit (Roche), following manufacturer's instructions. Plates were developed for 40 min in the dark, and the absorbance at 490 nm was quantified on a plate reader (PerkinElmer). Percent maximum LDH release was calculated for each strain relative to a 1% Triton X-100 lysis control. Statistical analysis was performed in Graphpad Prism v.9.2.0

### Epithelial barrier breach
16HBE-14o- cells were maintained in 1x MEM + 10% FBS. To establish semi-differentiated cultures, $2 \times 10^5$ cells were seeded onto 33 mm ThinCert Cell Culture Inserts with 8 µm pores (Greiner). Prior to seeding, inserts were coated with an extracellular matrix component mixture comprised of 10 µg/mL human fibronectin (Sigma), 100 µg/mL bovine serum albumin (Fisher), 30 µg/mL PureCol collagen (Sigma) in 1x MEM for 2 h at room temperature. Cultures were maintained with apical and basolateral media for 24 h, at which point the apical media was removed and 400 µL of MEM + 10% FBS was added to the basolateral chamber. Cells were allowed to differentiate at the air-liquid interface (ALI) for an additional 10 days, until reaching a transepithelial electrical resistance of ~500 Ω. The culture media was then replaced with 1x MEM in both the apical and basolateral chambers. ALI cultures were infected with 10 µL of bacteria ($5 \times 10^6$ colony forming units (CFUs), multiplicity of infection of 10 after growth for 20 h at 37 °C with 5% $CO_2$ and resuspended in HI). An additional 290 µL of 1x MEM was added to the apical chamber to accommodate a probe to monitor transepithelial resistance. At 30 min, 1 h, and 2 h post-infection, the basolateral media was collected and plated for CFUs. Statistical analysis was performed in Graphpad Prism v.9.2.0 and significant differences are shown.

### Juvenile rat infection model
*K. kingae* strains were swabbed from chocolate agar plates and resuspended in PBS to a final density of $5.0 \times 10^8$ CFU/mL. At least eight, 5-day-old Sprague–Dawley rat pups (Charles River Laboratories, Wilmington, MA) were injected via the intraperitoneal route with 0.1 mL of the bacterial resuspensions or PBS alone with a 27 1/2-gauge needle and then returned to their cage with a lactating dam. Animals were not sexed prior to infection. The experimental groups were housed separately and were monitored for signs of morbidity and mortality twice daily for a total of 5 days. Animals found to be moribund were euthanized by using $CO_2$ inhalation followed by secondary decapitation. The animal procedures were approved by the Children's Hospital of Philadelphia Institutional Animal Care and Use Committee (IACUC) under protocol IAC 19-001050. A Kaplan–Meier curve was

constructed, and statistical analyses were performed in Graphpad Prism v.9.2.0.

### Reporting summary

Further information on research design is available in the Nature Portfolio Reporting Summary linked to this article.

## Data availability

The sequencing data generated in this study have been deposited in the National Center for Biotechnology Information's (NCBI) Sequence Read Archive (SRA) and Genbank databases under BioProject PRJNA896475. The sequencing statistics and gene list data generated in this study are provided in the Supplementary Data 5–7. The files containing the phylogenetic trees in Newick format, and additional data, and the full statistical outputs from IQTree are available in a Github repository (https://github.com/danmorreale/Kingella-Phylogenomics). The VFDB and MEGARes databases are distributed as part of the Abricate distribution[53]. All other data have been deposited in a Github repository (https://doi.org/10.5281/zenodo.7970992)[69], and the source data are provided with this paper. Any other data and strains presented in this study are available from the authors by request. Source data are provided with this paper.

## Code availability

All custom scripts or programs used in the analyses presented here are available at the associated Github repository [https://github.com/danmorreale/Kingella-Phylogenomics].

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

## Acknowledgements

We want to thank Dr. Pablo Yagupsky for sharing his extensive strain collection with us, as well as Dr. Nataliya Balashova for generously

providing us with the stain PYKK081Δ*rtxA* (KKNB100). We would also like to thank all members of the St. Geme and Planet labs, particularly Dr. Nina Montoya and Kevin Hernandez for their constructive feedback and assistance during the preparation of this manuscript. This work was supported by the National Institute of Allergy and Infectious Diseases under 5T32AI141393 to D.P.M. as well as 1R01AI121015 and 1R01AI172841 to J.W.S.

## Author contributions

J.W.S. and P.J.P. secured funding for this work and supervised the study. D.P.M., J.W.S., and P.J.P. conceived of the study. D.P.M., E.A.P., and B.K.K. designed and performed in vitro and in vivo experiments and analyzed data. D.P.M. conducted in silico experiments. D.P.M., E.A.P., J.W.S., and P.J.P. interpreted the data. D.P.M. wrote the original draft of the manuscript. All authors edited and approved the final manuscript.

## Competing interests

The authors declare no competing interests.
