## [Peer Review File · Nature Communications]

Acquisition, co-option, and duplication of the rtx toxin system and the emergence of virulence in *Kingella*Reviewer #1 (Remarks to the Author):

RTX toxins are key virulence factors in many Gram-negative bacteria. The results presented in the manuscript advance our understanding of the pathogenesis of disease due to an understudied pathogen that is being recognized increasingly as an important source of morbidity in young children. There is substantial novelty in the findings and the study is significant for two major reasons. First, the work sheds light on the virulence and invasiveness of pediatric pathogen *Kingella kingae* and recently identified *Kingella negevensis*. In this manuscript the authors investigated and reported the relationship between amount of RtxA production and level of cytotoxicity. This manuscript also answers more general question on how some commensal organisms can acquire increased virulence, one of the ways of "evolutionary transition from commensal to pathogen". This could provide new insights into the family of RTX toxins and their mechanisms of action in killing host cells.

Suggestions to the authors:

1. Are there any examples of RTX genes multiplication in other bacteria? Please add to the Discussion.
2. Perform an analysis of outbreak *Kingella* strains for presence of *rtxA* duplications towards possible development of diagnostic tools
3. I would use term "cytotoxicity in vitro" instead of "virulence in vitro" in line 9.

Reviewer #2 (Remarks to the Author):

Morreale et al here describe the evolution of *K. kingae* and *K. negevensis* into human pathogens from their commensal ancestors. They describe the acquisition of the T1SS and *rtxA* toxin loci into both species, and the duplication of the toxin gene in *K. kingae*, as well as comparison to other bacterial species with homologous systems. They demonstrate that the duplication of *rtxA* can contribute to increased toxin production. Overall this study is a neat demonstration of how modern genomic approaches can be combined with wet-lab experimentation to obtain valuable insights into an otherwise under-studied pathogen.

Major comments

1. I have several comments about the tree in Figure 1b. First, why is the tree a cladogram and not a phylogram? The figure legend states it is a "phylogeny", but in a phylogeny the branch lengths represent substitutions per site, however in this tree all branch lengths are equal.
 - Can you please indicate on Fig 1b to which genomes are public and which genomes are from this study?
 - Why are there only three *K. negevensis* genomes in the tree? You mention in the paper that you sequenced an additional 12, why are these not included?
 - I don't find the bootstrap circles to be very helpful, it's difficult to assess sizes. I would recommend simply labelling nodes with <75% support with a circle and then writing the number on the branch, as there aren't very many
2. I was confused about why you needed to perform the mapping to *K. kingae* KWG-1 to prove that there was only a single copy of *rtxA* in *K. negevensis* that are arranged in a single locus (rather than two loci as in *K. kingae*). To me this should be clear simply from the annotations of the complete genome assemblies and a BLAST for *rtxA*? Did you have Illumina data as well for these 12 strains, which would allow you to do a complete assembly using a hybrid approach (as described by Wick et al, "Tricycler: consensus long-read assemblies for bacterial genomes" in *Genome Biology*, 2021), enabling more accurate annotation?
 - You mention there are structural rearrangements in *K. kingae* compared to *K. negevensis* – I think it might be helpful to add a panel to Figure 2 that shows the locus in *K. negevensis* with its BLAST comparison it to the loci in *K. kingae* using a genoplots or ACT figure?
3. Line 399: You mention in the discussion you think the likely donor of the toxin region is M.

bovis. It took me a long time to realise that you meant *Moraxella bovis*, not *Mycobacterium bovis*, as I don't believe you define *M bovis* at first use. I think it would be valuable to summarise again in the discussion at this point all the lines of evidence that make you hypothesise this – from reading I think it's a combination of %GC and the relatedness of RtxA etc to the *Moraxella* group. Is there any reason why you think the donor is *M. bovis* in particular, and not the other species?

4. Please provide individual accessions (either read or sample) for all genomes in Table S1. In Table S2 it would be useful to provide the assembly accessions for each genome.
- Were the long reads uploaded? Or at least the completed genome assemblies?
- It would be useful to add a column to Table S1 indicating the geographic origin of each strain, if that information is able to be shared.

Minor comments

5. Line 32 – what is "the mobile genetic element"? should it be "a"? or is there a specific element, in which case it should be named?

6. Line 97 – copy number of what? rtxA?

7. Are you able to identify more specifically the transposition genes, shown in teal in Fig 2? The paper says they belong to the IS5 family, but what insertion sequences/transposons are they? Are these IS commonly found in the *Moraxella* group? If so, it would be helpful to label them on the figure.

Reviewer #3 (Remarks to the Author):

The authors have studied the rtx gene cluster(s) in pathogenic species of *Kingella*. They suggest heterologous origins for different components and hypothesize that rtxA copy number might be an important determinant of pathogenicity.

Major concerns:

While the study has generated a significant amount of sequence and other data, I am not completely convinced that the work represents a significant advance on the current state of understanding, or that all assertions are rigorously justified.

1) lines 244-245, to state an absence of *K. negevensis* genome data (and to cite Opota, O. et al 2017) is misleading. These authors presented an almost complete *k. negivensis* genome and illustrate the layout of the RTX locus, comparing to the 2 arrangements found in *L. Kingae*. I find figures 2B and C to be difficult to interpret. They would be clearer with more explicit legends.

2) No details of the minion assemblies generated in this study (or accession numbers) appear to be provided

3) I have considerable difficulty with reconciling the hypotheses regarding the evolutionary origin of the *Kingella* rtx genes with the trees presented in the supplementary data. By my interpretation, for all of these genes (including tolC), *Moraxella* homologs fall in positions completely consistent with "recent" common ancestry of the whole *Kingella* and *Moraxella* rtx complement. These trees should all be labelled with bootstrap support values, and I would suggest constrained alternative topology testing (SH or approximately unbiased tests) to examine hypotheses of HGT. Overall I do not find the evidence for reconstitution of the complete rtx system from a mix of xenologous and "host" components compelling.

4) authors hypothesis that in the MRA of type I and type II *kingella kingae* genotypes, the ancestral rtx locus was split into two loci and that in type II isolates a duplication of two genes

from locus B has allowed the reconstitution of the ancestral locus A. A more simple explanation might be that the first event might be the DUPLICATION of locus B genes (in the k. Kingae ancestor), followed by LOSS of two genes from locus A in some genotypes. Importantly, this scenario posits 2 RTXa copies as "ancestral" in k. kingae genotype 2 and not derived. While on the tree of strains it is one step less parsimonious than the presented model, it does not require a specific site of reintegration of duplicated genes. Can the authors refute this scenario? would it make a material difference to hypotheses, data interpretation etc in the work?

4) Line 320. The p-value that suggests an "enrichment" of invasive strains among genotype II isolates is rather marginal. This is fundamental as it is the justification for the further functional analyses of RTXa copy number on pathogenicity. Can the authors show that there is not a significant geographical or time of isolation bias associated with genotype II. This is relevant because of potential sampling biases associated with localities, outbreaks, historical sampling policies.

5) Lines 331-337. I presume that the authors refer to a presence/absence GWAS on accessory genome genes? Did they consider a classical GWAS on SNP/indel/repeat number/ gene mutational burden etc. variants?

7) line 416-417 "Reconstitution of a complete RTX system by this mechanism is an essential and defining feature in the evolution of invasive disease by this genus." I am not convinced that compelling evidence for this chimeric origin of the system has been presented.

Minor comments:

1) At least 2 references are present twice in the bibliography (Hess, J. F 2006, Opota, O. et al 2017 check carefully throughout please)

2) for all phylogenetic analyses, the number of sites included in alignments used for reconstruction should be made available (These may be present in additional materials, but I was unable to access these for review).

3) line 425... I am confused here... do the authors mean "in-vivo"?

4) There is some minor repetition throughout the manuscript, also, in a few parts, the order of arguments could probably be revised to increase clarity.

Thank you for your review of our manuscript entitled “Acquisition, co-option, and duplication of the *rtx* toxin system and the emergence of virulence in *Kingella*,” authored by Daniel P. Morreale, Eric A. Porsch, Brad K. Kern, Joseph W. St. Geme, III, and myself (NCOMMS-22-50127-T). We greatly appreciate the comments from the reviewers and believe that these comments have enabled us to strengthen the manuscript. Our response includes additional bioinformatic analysis, additional whole genome sequencing, and new interpretation of results throughout. Our responses to the comments can be found in bold below each reviewer comment.

In addition to the changes noted associated with the reviewer comments, we have modified the following sections to encompass the additional experiments and analyses performed in the process of preparing the revised manuscript:

- Figure 1B has been replaced and reformatted as described below.
- Figure 2A reflects additional whole genome sequencing done for this resubmission, and this replaces the original Figure 2A-C
- Whole genome sequences of *K. negevensis* are in the PGAP pipeline for public release. Those with available accession numbers have been updated in the supplemental material for this manuscript.

All modifications in the manuscript are highlighted in yellow.

I am hopeful that our revised manuscript is now acceptable for publication in *Nature Communication*. Please let me know if you have any questions or need additional information.

Sincerely,

Paul J. Planet, MD, PhD
Assistant Professor of Pediatrics
University of Pennsylvania
Children’s Hospital of Philadelphia

Reviewer #1:

1. Are there any examples of RTX genes multiplication in other bacteria? Please add to the Discussion

While uncommon, there is at least one other example of possible *rtx* gene multiplication among Gram-negative bacteria. In particular, in *Actinobacillus pleuropneumoniae* five functionally distinct and differentially regulated related toxins have been described (encoded by *apxI*, *apxII*, *apxIII*, *apxIV*, and *apxV*). However, the evolutionary history of these toxins and the prevalence of toxin gene copy number variability in the population structure are unclear. In the original manuscript, we mentioned this example in the Discussion, but in the revised manuscript we have added more detail.

We thank the reviewer for prompting us to explore this important system more thoroughly, highlighting a system in which toxin genes are expressed from different loci than the TISS genes. We believe that this example provides additional evidence that the *rtxCrtxAtoIC* locus could have been acquired independently of the *rtxB* and *rtxD* genes.

2. Perform an analysis of outbreak *Kingella* strains for presence of *rtxA* duplications towards possible development of diagnostic tools

We agree that *rtxA* is a tempting target for a diagnostic tool to identify potentially more virulent *K. kingae* strains and develop approach to prevent disease. Others have developed a tool that takes a similar approach (Lehours et al 2011), with some success, with the expected caveat that tools targeting *rtxA* cannot differentiate between *K. kingae* and *K. negevensis* (Houmami et al. 2017). Critical to the work presented here, a consideration of *rtxA* copy number in a prospective clinical cohort would be an excellent study to further test the hypothesis that copy number impacts virulence. Considering that infection with *K. kingae* is still a relatively infrequent event (approximately 5-10 cases per year at our large children's hospital), a definitive clinical study is beyond the scope of this work. It is also notable that we expect to see some isolates with a single copy of *rtxA* that cause invasive disease.

3. I would use term "cytotoxicity in vitro" instead of "virulence in vitro" in line 9.

We have made this suggested change in the revised manuscript.

Reviewer #2:

Major comments

1. I have several comments about the tree in Figure 1b. First, why is the tree a cladogram and not a phylogram? The figure legend states it is a "phylogeny", but in a phylogeny the branch lengths represent substitutions per site, however in this tree all branch lengths are equal.

A cladogram was selected to better show branch length to the leaves. However, given that this figure predominantly depicts species delineations, we agree that a phylogram would be better suited. In the revised manuscript, we have modified the figure accordingly. In this new figure, it is difficult to resolve clade boundaries due to the tree scale, and thus we have also added a newick formatted file in the supplementary materials. To further aid in visibility of the branch length, we modified the metadata plotted on this dendrogram.

- Can you please indicate on Fig 1b to which genomes are public and which genomes are from this study?

We have modified this tree to primarily include isolates that were newly sequenced in this study, along with at least one representative strain that is publicly available as "signpost" isolates. Publicly available isolates are bolded. The figure legend has been modified to reflect these changes.

- Why are there only three *K. negevensis* genomes in the tree? You mention in the paper that you sequenced an additional 12, why are these not included?

Originally, *K. negevensis* isolates used in this study were sequenced only with MinION long reads, aiming to allow assessment of *rtxA* copy number, rather than to generate extremely high-quality, circular genomes. MinION sequencing can have base-error rates as high as 10%, and base-calling is typically corrected using Illumina sequencing reads. Given that Figure 1B relies on a SNP matrix generated from a whole genome alignment, we felt it was inappropriate to include these isolates, as they may have introduced significant noise into the dataset, without improving our understanding of virulence determinant conservation. However, to complete this work, we decided to perform additional sequencing to improve the quality of these genomes. In the revised manuscript, we have included all 12 isolates of *K. negevensis* in the alignment and in Figure 1B.

- I don't find the bootstrap circles to be very helpful, it's difficult to assess sizes. I would recommend simply labelling nodes with <75% support with a circle and then writing the number on the branch, as there aren't very many.

We have made the suggested change in the revised manuscript.

2. I was confused about why you needed to perform the mapping to *K. kingae* KWG-1 to prove that there was only a single copy of *rtxA* in *K. negevensis* that are arranged in a single locus (rather than two loci as in *K. kingae*). To me this should be clear simply from the annotations of the complete genome assemblies and a BLAST for *rtxA*? Did you have Illumina data as well for these 12 strains, which would allow you to do a complete assembly using a hybrid approach (as described by Wick et al, "Tricycler: consensus long-read assemblies for bacterial genomes" in Genome Biology, 2021), enabling more accurate annotation?

- You mention there are structural rearrangements in *K. kingae* compared to *K. negevensis* – I think it might be helpful to add a panel to Figure 2 that shows the locus in *K. negevensis* with its BLAST comparison to the loci in *K. kingae* using a genoplots or ACT figure?

As noted above, we have performed additional Illumina sequencing on these *K. negevensis* isolates. As suggested, these were then assembled using a hybrid assembler, which yielded exceptionally high-quality genomes. These genomes have been submitted to NCBI for publication. We have created a new Figure 2 showing BLAST similarity between the locus in *K. kingae* KWG-1, ATCC23330, and these *K. negevensis* isolates in lieu of reporting read depth. Using this new information, we have also modified the Methods section and the Results section.

3. Line 399: You mention in the discussion you think the likely donor of the toxin region is *M. bovis*. It took me a long time to realise that you meant *Moraxella bovis*, not *Mycobacterium bovis*, as I don't believe you define *M. bovis* at first use. I think it would be valuable to summarise again in the discussion at this point all the lines of evidence that make you hypothesise this – from reading I think it's a combination of %GC and the relatedness of *RtxA* etc to the *Moraxella* group. Is there any reason why you think the donor is *M. bovis* in particular, and not the other species?

We have now performed additional analyses using gene phylogenies to clarify the relationship between the *rtx* locus in *Moraxella bovis* the pathogenic *Kingella* species. We have modified the Results and Discussion to suggest the hypothesis that the locus was acquired in *M. bovis* from a *Kingella* ancestor, not the other way around. was acquired from *Moraxella bovis*.

4. Please provide individual accessions (either read or sample) for all genomes in Table S1. In Table S2 it would be useful to provide the assembly accessions for each genome.

- Were the long reads uploaded? Or at least the completed genome assemblies?

- It would be useful to add a column to Table S1 indicating the geographic origin of each strain, if that information is able to be shared

We have modified these tables to include the requested information. All reads have been submitted to SRA. Where available, we have included the geographic origin of isolates.

Minor comments

5. Line 32 – what is “the mobile genetic element”? should it be “a”? or is there a specific element, in which case it should be named?

We have clarified this language in the revised manuscript. This is the first study to propose a specific element in this detail.

6. Line 97 – copy number of what? rtxA?

We have clarified the language in this section and have made additional changes to the methods detailed in this section to reflect experiments by the reviewers.

7. Are you able to identify more specifically the transposition genes, shown in teal in Fig 2? The paper says they belong to the IS5 family, but what insertion sequences/transposons are they? Are these IS commonly found in the *Moraxella* group? If so, it would be helpful to label them on the figure.

IS5 family transposable elements are found in an extremely wide range of bacteria and archaea. We attempted to better classify these sequences into one of the subgroups of the IS5 family, but each of these putative genes appears to be only a fragment of larger transposon genes, and we were unable to reliably classify them to IS5 subgroups. We hypothesize that these genomic sites are recombination hot-spots, with a series of integrations and excisions resulting in the fragments that we observe.

Reviewer #3 (Remarks to the Author):

1) lines 244-245, to state an absence of *K. negevensis* genome data (and to cite Opota, O. et al 2017) is misleading. These authors presented an almost complete *K. negevensis* genome and illustrate the layout of the RTX locus, comparing to the 2 arrangements found in *K. kingae*. I find figures 2B and C to be difficult to interpret. They would be clearer with more explicit legends.

We have rephrased this section to better acknowledge the genome and RTX locus published by Opota et al.

In the revised manuscript, we have removed Figures 2B and C and replaced them with more complete and easier to interpret diagrams (Fig 2A).

2) No details of the minion assemblies generated in this study (or accession numbers) appear to be provided

We apologize for this oversight. As stated in our response to Reviewer 2, we have corrected the accession number information and the associated metadata tables.

3) I have considerable difficulty with reconciling the hypotheses regarding the evolutionary origin of the *Kingella* rtx genes with the trees presented in the supplementary data. By my interpretation, for all of these genes (including tolC), *Moraxella* homologs fall in positions completely consistent with "recent" common ancestry of the whole *Kingella* and *Moraxella* rtx complement. These trees should all be labelled with bootstrap support values, and I would suggest constrained alternative topology testing (SH or approximately unbiased tests) to examine hypotheses of HGT. Overall I do not find the evidence for reconstitution of the complete rtx system from a mix of xenologous and "host" components compelling.

We have made several changes to this analysis to better address the concerns raised by the reviewer in this comment. First, we have reconstructed our phylogenetic trees using IQTree to include a wider breadth of species. We revised the manuscript to reflect this change, and to include bootstrap values on these trees (Fig. S2). This analysis allowed us to clearly represent the position of *Moraxella* species in our tree and revise our interpretation of the origins of the rtx locus. Additionally, we have included additional statistical analyses (SH, KW, and AU tests) to test congruency between the RtxA, RtxD, and RtxB trees.

Careful examination of the gene trees also led us to identify a close relative of *Kingella rtxA* in *Actinobacillus* species that is found in a locus that lacks homologs of *rtxB* and *rtxD*. This observation further supports the possibility that the *rtxCrtxAtoIC* locus could have been acquired independently.

4) The authors hypothesis that in the MRA of type I and type II *Kingella kingae* genotypes, the ancestral rtx locus was split into two loci and that in type II isolates a duplication of two genes from locus B has allowed the reconstitution of the ancestral locus A. A more simple explanation might be that the first event might be the DUPLICATION of locus B genes (in the *K. kingae* ancestor), followed by LOSS of two genes from locus A in some genotypes. Importantly, this scenario posits 2 RTXa copies as "ancestral" in *K. kingae* genotype 2 and not derived. While on the tree of strains it is one step less parsimonious than the presented model, it does not require a specific site of reintegration of duplicated genes. Can the authors refute this scenario? would it make a material difference to hypotheses, data interpretation etc in the work?

Initially, we also favored the scenario suggested by the reviewer. However, this conclusion is not supported by the tree topology. In our phylogeny rooted by an outgroup (here *K. negevensis*), we always find multiple clades/branches that have a single copy of *rtxA* that are basal to derived clades that have 2 copies of *rtxA*. This pattern occurs at the base of the tree leading into the clade where genotype II predominates (red clade II), but it also occurs in other clades. For instance, we occasionally observe genotype II isolates in clades with primarily genotype I isolates (e.g. AA392C5, KK200, AA478). Based on ancestral reconstruction, these are all strongly predicted to have been derived duplications rather than losses. Of course, we cannot completely rule out the scenario proposed by the reviewer.

Additionally, while preparing the isogenic mutants used in this manuscript, we experienced targeted mutations in locus B recombining into locus A very frequently, suggesting that site specific recombination between these two loci is both (1) frequent and (2) rapid.

4) Line 320. The p-value that suggests an "enrichment" of invasive strains among genotype II isolates is rather marginal. This is fundamental as it is the justification for the further functional analyses of RTXa copy number

on pathogenicity. Can the authors show that there is not a significant geographical or time of isolation bias associated with genotype II. This is relevant because of potential sampling biases associated with localities, outbreaks, historical sampling policies.

We agree that sampling bias could skew this association. The majority of the isolates used in this study and the majority of available *K. kingae* isolates were recovered by Dr. Pablo Yagupsky in southern Israel over the course of approximately 20 years. Prior to Dr. Yagupsky's work, which included samples from both healthy carriers and children with disease, *K. kingae* was rarely cultured from clinical specimens. Therefore, it is very difficult to assess time as a confounding factor. Where possible, we have included strains that are historical (e.g. ATCC23330 was recovered in the early 1960's, and ATCC23331 and ATCC23332 were recovered in the early 1970's), and global (e.g. KK01 and KK03 were recovered in the United States, and KWG-1 was recovered in France). We identify several genotype II isolates that are historical and/or from the global collection, including the strains listed above with the exception of ATCC23330. In the revised manuscript, we have added a comment on this topic in the Discussion.

5) Lines 331-337. I presume that the authors refer to a presence/absence GWAS on accessory genome genes? Did they consider a classical GWAS on SNP/indel/repeat number/ gene mutational burden etc. variants?

We chose to focus this study on gene presence/absence rather than a more classical, SNP-based GWAS. We did consider SNP-based GWAS, but initial explorations showed very high numbers of SNPs across the *Kingella kingae* phylogeny (ranging from >5 to >10,000 SNPs for the sequences used in this study). Experience with this type of variability in the past has suggested that the signal to noise ratio may preclude reliable conclusions. Therefore, we did not pursue this analysis further.

7) line 416-417 "Reconstitution of a complete RTX system by this mechanism is an essential and defining feature in the evolution of invasive disease by this genus." I am not convinced that compelling evidence for this chimeric origin of the system has been presented.

We have softened this statement to read "Given our virulence model and genomic data, we suggest that reconstitution of a complete RTX system by this mechanism is an essential and defining feature in the evolution of invasive disease by this genus." We have now offered additional evidence for a chimeric origin based on taxonomic distribution, gene tree phylogeny and congruence, and the existence in close relatives of loci in which the secretion system is found in a different locus from the toxin encoding genes.

Minor comments:

1) At least 2 references are present twice in the bibliography (Hess, J. F 2006, Opota, O. et al 2017 check carefully throughout please)

We have removed the repeated references.

2) for all phylogenetic analyses, the number of sites included in alignments used for reconstruction should be made available (These may be present in additional materials, but I was unable to access these for review).

We have amended the Results sections to include the number of variable and invariant sites found in each alignment. While the full alignments are available with the additional data, we agree that this information is important to include in the body of the manuscript.

3) line 425... I am confused here... do the authors mean "in-vivo"?

We have amended the language in this section and added additional discussion to better contextualize our results.

4) There is some minor repetition throughout the manuscript, also, in a few parts, the order of arguments could probably be revised to increase clarity.

We have made an effort to remove unnecessary repetition in the revised manuscript.

Reviewer #2 (Remarks to the Author):

I am happy with the changes the authors have made to the manuscript. In particular Figure 1 is much clearer now. However, could the authors please include the accessions of all the public genomes they used in this study into Table S1 (either reads, assemblies or biosamples?) This will aid with reproducibility.

Reviewer #3 (Remarks to the Author):

My concerns have all been adequately addressed. I think that the current version of the manuscript is much clearer and the conclusions either better supported or more suitably qualified as appropriate